# Acyl-CoA-Binding Domain-Containing 3 (ACBD3; PAP7; GCP60): A Multi-Functional Membrane Domain Organizer

**DOI:** 10.3390/ijms20082028

**Published:** 2019-04-24

**Authors:** Xihua Yue, Yi Qian, Bopil Gim, Intaek Lee

**Affiliations:** 1School of Life Science and Technology, ShanghaiTech University, Pudong, Shanghai 201210, China; yuexh@shanghaitech.edu.cn (X.Y.); qianyi@shanghaitech.edu.cn (Y.Q.); 2School of Physical Science and Technology, ShanghaiTech University, Pudong, Shanghai 201210, China; bgim@shanghaitech.edu.cn

**Keywords:** ACBD3, Golgi, PAP7, GCP60, PI4KB

## Abstract

Acyl-CoA-binding domain-containing 3 (ACBD3) is a multi-functional scaffolding protein, which has been associated with a diverse array of cellular functions, including steroidogenesis, embryogenesis, neurogenesis, Huntington’s disease (HD), membrane trafficking, and viral/bacterial proliferation in infected host cells. In this review, we aim to give a timely overview of recent findings on this protein, including its emerging role in membrane domain organization at the Golgi and the mitochondria. We hope that this review provides readers with useful insights on how ACBD3 may contribute to membrane domain organization along the secretory pathway and on the cytoplasmic surface of intracellular organelles, which influence many important physiological and pathophysiological processes in mammalian cells.

## 1. Introduction

The Golgi apparatus has traditionally been considered as a central sorting station along the secretory pathway for newly synthesized secretory proteins [1,2]. More recently, however, many studies have described its roles in intracellular signaling and feedback mechanism that leads to more integrative cellular decision-making for cell division, differentiation, apoptosis, and sensing of cellular secretory activity [3,4,5]. 

Acyl-CoA-binding domain-containing 3 (ACBD3) was originally found in the early 2000s as a Golgi scaffolding protein that helped the function of Golgin tethers, such as Giantin and Golgin-160 in membrane trafficking and apoptosis [6,7,8,9]. ACBD3 is known to directly bind palmitoyl-CoA using its ACBD domain, while its GOLD (Golgi Dynamics) domain mediates ACBD3 targeting to the Golgi and other membranes (Figure 1) [10,11]. Over the past 20 years, highly diverse interaction partners of ACBD3 have been discovered, including PI4KB, Golgin45, TBC1D22, PPM1L, DMT1, PKA, FAPP2, TUG, PARP-1, *mHtt*, Numb, as well as several bacterial and viral proteins [10,12,13,14,15,16,17,18,19,20,21]. 

In most of these cases (except for Numb), ACBD3 seems to function as a part of a multi-protein complex and is likely to form functional membrane micro-domains at the Golgi and mitochondria. For example, ACBD3 has been shown to function as a Golgi docking site for PI4KB, which plays crucial roles for transport carrier formation at the Golgi [22]. Interestingly, ACBD3–PI4KB interaction appears to be linked to the perinuclear vacuole/sites, proximal to the Golgi, where many viruses and bacteria promote their intracellular proliferation by hijacking and exploiting membrane trafficking machineries in the host cells [12,19,22,23,24,25,26]. 

In this article, we discuss recent discoveries on novel ACBD3 interaction partners, through which membrane domain organization emerges as a basic theme for highly diverse physiological and pathophysiological roles played by ACBD3.

## 2. ACBD3 in Steroidogenesis

Liu et al. identified ACBD3 (PAP7: PBR and PKA-associated protein 7) in a yeast two-hybrid experiment, screening for PBR (peripheral-type benzodiazepine receptor) and PKA (protein kinase A)-interacting proteins [27]. PBR, a cholesterol binding protein, has been shown to mainly localize in the outer membrane of the mitochondria and participate in steroid biosynthesis by accelerating the transport of cholesterol from the outer to the inner membrane of mitochondria. PKA has also been demonstrated to be involved in steroid biosynthesis upon activation by increased concentration of cyclic AMP. PKA is a heterotetramer with two catalytic (C) subunits and two regulatory (R) subunits, which consist of four different types, namely, RIα, RIβ, RIIα, and RIIβ. PKA holoenzyme is auto-inhibitory since the C subunit is inactivated by its binding to the R subunit. Once cyclic AMP binds to the R subunit, the C subunit is released and activated. Purified recombinant GST-tagged ACBD3 fragments 228-445 (linker between the ACBP and GOLD domains plus the GOLD domain) and 212-369 (linker between the ACBP and GOLD domains plus the first half of the GOLD domain) were able to pull down PBR from mitochondrial extracts and the recombinant regulatory subunit of PKA in GST-pulldown assays, respectively. 

Overexpression of the full-length ACBD3 in MA-10 Leydig cells greatly improved steroid synthesis upon stimulation by hCG, whereas lower expression of ACBD3 by antisense oligonucleotides specific to ACBD3 inhibited the steroidogenesis by MA-10 cells. Interestingly, overexpression of the fragments 228-445 and 212-369 of ACBD3, which contain the PBR and PKA binding motifs respectively, substantially reduced the progesterone production stimulated by hCG [28]. These fragments may compete with the endogenous ACBD3 and inhibit PBR or PKA binding to endogenous ACBD3, therefore preventing cholesterol transport to mitochondria mediated by the PBR–PKA–ACBD3 complex. 

## 3. ACBD3 in Lipid Metabolism and Metabolic Homoeostasis

While ACBD3 is a positive regulator of steroidogenesis, it seems to negatively regulate de novo cholesterol synthesis through inhibiting SREBP1 (sterol regulatory element-binding protein 1) [29]. Chen et al. showed that ACBD3 was a new binding partner and maturation modulator of SREBP1. ACBD3 inhibited SREBP1-sensitive promoter activity of fatty acid synthase (FASN). Moreover, ACBD3 blocked intracellular maturation of SREBP1 probably through directly binding with the lipid regulator rather than disrupting SREBP1–SCAP–Insig1 interaction. ACBD3 overexpression inhibited de novo fatty acid synthesis by regulating FASN and ACC (acetyl-CoA carboxylase). These observations extended the regulatory properties of ACBD3 to fatty acid biogenesis beyond cholesterol biogenesis and steroidogenesis [29]. 

Later studies by the same group have shown that ACBD3 was involved in modulating NAD^+^ metabolism through activating poly(ADP-ribose) polymerase 1 (PARP1) [21]. PARP1 is a well-characterized stress response protein, which mediates various DNA repair pathways and maintains genomic stability. PARP1 also plays a regulatory role in metabolism and lipid metabolism [30]. They showed that overexpressed ACBD3 significantly reduced cellular NAD^+^ content via enhancing PARP1’s polymerase activity and enhancing auto-modification of the enzyme in a DNA damage-independent manner. Furthermore, they found that enhanced ERK1/2 activity (extracellular signal-regulated kinase1/2) and inhibited SREBP1-controlled fatty acid biosynthesis play crucial roles in this ACBD3-regulated PARP1 activation. Importantly, oxidative stress-induced PARP1 activation is greatly attenuated by knocking down the ACBD3 gene. These findings broaden our views on the biological roles played by ACBD3 and may provide novel insights into the mechanisms underlying how lipid-binding proteins affect metabolic homoeostasis via PARP1-involved signaling pathway.

## 4. ACBD3 in Regulation of Ceramide and Glucosylceramide Transport

Glycosphingolipids (GSLs) are important components of the plasma membrane (PM), with key roles in cell signaling, adhesion, proliferation, and differentiation. GSLs are synthesized at the Golgi complex from glucosylceramide (GlcCer), which is synthesized from ceramide at the cytosolic leaflet of early Golgi membranes. Upon translocation to the luminal leaflet, GlcCer is galactosylated to lactosylceramide (LacCer), which can then be converted into complex GSLs in later Golgi compartments [31,32]. GlcCer can be transported through the Golgi complex via membrane trafficking and via non-vesicular transfer owing to the action of the cytosolic GlcCer transfer protein FAPP2, which fosters GSL synthesis [33,34]. 

PPM1L, an endoplasmic reticulum (ER)-resident transmembrane protein phosphatase, appears to be involved in the regulation of ceramide trafficking at ER–Golgi membrane contact sites [35]. ACBD3 was shown to play a crucial role in recruiting PPM1L to ER–Golgi membrane contact sites to dephosphorylate the ceramide transport protein CERT and regulate ceramide transport [17]. More recently, ACBD3 was identified as a FAPP2-interacting partner [9]. This interaction may facilitate the recruitment of FAPP2 to ER–Golgi membrane contact sites and translocate GlcCer to the lumen of ER. GlcCer in the ER is then transported to the Golgi cisternae via vesicle trafficking for GSL biosynthesis. These studies revealed critical roles for ACBD3 in regulating cellular sphingolipid metabolism. Since ACBD3 is important in maintaining the integrity of Golgi morphology, these data also establish the importance of the Golgi complex for the transfer of GlcCer and complex GSL synthesis.

## 5. ACBD3 in Golgi Structure Maintenance

As mentioned in the introduction, ACBD3 was originally described as a Golgi scaffolding protein, which interacts with Golgin tethers, such as Giantin [6]. Recently, through proximity-based in vivo tagging and using a proteomics approach, we identified ACBD3 as a novel binding partner of Golgin45, a medial Golgi-localized Golgi structural protein [13]. ACBD3 appears to use its GOLD domain to interact with Golgin45 and enhance Golgin45 targeting to the Golgi. Previous studies have indicated that ACBD3 interacts with at least three other membrane trafficking proteins at the Golgi, including Giantin, Golgin-160, and TBC1D22, a Rab33b-GTPase activating protein (GAP) [6,7,16]. 

Unexpectedly, however, Giantin and Golgin-160 were not a part of the Golgin45–ACBD3 complex, whereas TBC1D22 was strongly detected during reciprocal co-immunoprecipitation experiments. In addition, GRASP55, a well-known Golgin45-binding partner and Golgi stacking protein, was also selectively enriched in the Golgin45–ACBD3–TBC1D22 complex. Thus, this large multi-protein complex (GRASP55–Golgin45–ACBD3–TBC1D22) seems to form a membrane micro-domain between the medial-Golgi cisternae and contributes to membrane trafficking and Golgi structure maintenance. As Rab33b had been described to be a part of Rab1 (*cis*)–Rab33b (*medial*)–Rab6 (*trans*) cascade in the mammalian Golgi apparatus [36,37], it is intriguing to speculate that this multi-protein complex may also play an integral role in regulating membrane identity of the medial Golgi cisternae. Further study is required to probe the validity of this hypothesis.

## 6. ACBD3 in Iron Uptake

At the mitochondria, stimulation of NMDAR (N-methyl-D-aspartate receptor) leads to activation of nNOS (neuron nitric oxide synthase), which generates NO (nitric oxide) and activates Dextras1 (dexamethasone-induced Ras-related protein 1) by its S-nitrosylation on cysteine 11, and results in iron uptake through Dextras1′s binding proteins, PAP7 and DMT1 (the divalent metal transporter 1) [18,38]. In undifferentiated PC12 cells, Dexras1 was found to bind to PAP7 in yeast two-hybrid, GST-pulldown and co-immunoprecipitation assays. 

The Dexras1-interacting site on PAP7 has been shown to be between amino acids 193-444, composed of the linker connecting the ACBP and the GOLD domains, and the first half of the GOLD domain as well. When undifferentiated PC12 cells were treated with different NO donors, that is, GSNO (S-nitrosoglutathione), SNP (sodium nitroprusside), or DETA NONOate, Dextras1 was S-nitrosylated and NTBI (non-transferrin bound iron) uptake was enhanced in a NO donor–concentration dependent manner. In mouse primary cortical neuron cells, activation of NMDAR by treatment of NMDA increased NTBI uptake, which was inhibited by preincubation of NMDAR antagonist or nNOS gene knockout [39].

## 7. ACBD3 Is Exploited by Viral and Bacterial Proteins to Promote Their Replication

ACBD3 was identified in a mammalian two-hybrid assay as a host protein that utilizes its C-terminal GOLD region to bind the Aichi virus nonstructural proteins 2B, 2C, and 3A [26]. Four hours after infection, ACBD3 was found to colocalize with 2B, 2C, and 3A in clusters in the cytoplasm, whereas other Golgi proteins, namely, Giantin, GM130, and TGN46, were not distributed with ACBD3. 

More recently, ACBD3 has also been shown to interact with the nonstructural protein 3A protein of plus-strand RNA viruses (picornaviruses), including human rhinovirus 14, coxsachievirus B3, and poliovirus [12,19,22,25,26,40,41,42,43,44,45,46,47]. The 3A protein is required for reorganization of the Golgi and replication of virus in host cells. In 293T cells, the Golgi proteins ACBD3 and PI4KB have been demonstrated to co-immunoprecipitate with 3A proteins through their N-terminus. Knockdown with siRNA specific to GBF1 (Golgi-specific brefeldin A resistance guanine nucleotide exchange factor 1), PI4KB, or ACBD3 greatly reduced, if not completely abolished, the replication of poliovirus and Aichi virus in HeLa cells. Mutations of the PI4KB-interacting amino acids in the N-terminus of 3A decelerated virus replication by insufficient recruitment of PI4KB [19].

ACBD3 was also demonstrated to be important for bacteria replication by presenting bacterial effector proteins close to the Golgi in host cells. ACBD3 was found to interact with bacterial protein SseG and SseF from *Salmonella typhimurium* in yeast two-hybrid assays and by co-immunoprecipitation in HeLa cells, but the interaction between SseF and ACBD3 is dependent on SseG. When ACBD3 was knocked down by siRNA in HeLa cells, the association of wild-type *Salmonella typhimurium* with the Golgi network was reduced, similar to the effect in cells infected by *S. typhimurium* carrying a null mutation in SseG. The bacterial replication was defective in HeLa cells infected by bacteria carrying both of the non-ACBD3-interacting mutant genes, *sseG^S67G^* and *sseF^C109W,D129G^*. However, when either *sseG^S67G^* or *sseF^C109W,D129G^* was expressed individually by *S. typhimurium* in HeLa cells, the association of *Salmonella*-containing vacuoles (SCV) and the Golgi in host cells was intact.

## 8. ACBD3 in Intracellular Retention of GLUT4 Storage Vesicles (GSV)

Insulin-induced translocation of glucose transporter is an important mechanism that controls blood glucose level, and any disruption in this process may lead to insulin resistance and diabetes. Belman et al. have recently shown that GSV retention in unstimulated adipocytes requires ACBD3 and overexpressed ACBD3 further enhances insulin-responsive GLUT4 translocation to the plasma membranes [14]. This effect was mediated by Sirt2-dependent acetylation of TUG (Tether containing a UBX domain for GLUT4), which in turn regulates TUG interaction with Golgi associated proteins, including ACBD3, PIST, and Golgin-160. It is interesting to note that Sirt2 targets FoxO1 and PGC1-α as well as Akt, thereby potentially affecting adipocyte differentiation and insulin-dependent signaling.

## 9. ACBD3 in Huntington’s Disease (HD)

Huntington’s disease is a neurodegenerative disease, caused by mutations in the gene encoding the huntingtin protein (*Htt*), leading to polyglutamine repeats expansion and the resulting cytotoxicity [48]. Sbodio et al. reported that ACBD3 is markedly increased in HD patients and interacts with mutant *Htt* (*mHtt*) and a small GTPase, Rhes (Ras homolog enriched in striatum) at the Golgi to form ACBD3/*mHtt*/Rhes complex [15]. This protein complex was found to promote *mHtt*/Rhes-dependent cytotoxicity [49], as depletion of ACBD3 entirely abolished *mHtt*/Rhes cytotoxicity, while its overexpression enhanced the cytotoxicity [15]. 

Strikingly, the authors also found that ACBD3 level was significantly increased by stress-inducing agents for the Golgi or the ER or the mitochondria, respectively, suggesting that prolonged cellular stress and up-regulation of ACBD3 level may be one of the causes for the cytotoxicity and onset of Huntington’s disease.

## 10. ACBD3 in Cancer

Lastly, but importantly, it has been reported that ACBD3 is up-regulated in breast cancer and associated with advanced pathoclinical features as well as poor prognosis in breast cancer [50]. In the study by Huang et al., overexpressing ACBD3 promoted, whereas silencing ACBD3 inhibited self-renewal of breast cancer cells in vitro. Furthermore, the tumorigenicity of breast cancer cells in vivo was also found to be increased significantly. 

It was shown that up-regulating ACBD3 promoted the self-renewal and tumorigenesis of breast cancer cells via activating the Wnt/beta-catenin signaling pathway [50]. These findings seem to suggest a novel ACBD3-dependent regulatory mechanism in breast cancer.

There is emerging evidence that the capacity for self-renewal is dysregulated in cancer stem cells (CSCs) and that CSCs also possess properties that make them resistant to chemotherapy and radiation [51,52]. Notably, the expression of ACBD3 was found to be 3.8-fold higher in non-responders (PD) than in responders (PR) to gefitinib in the response of lung cancer cells to gefitinib [53]. As previously reported, ACBD3 also regulates Numb signaling during asymmetric cell division in neural progenitor cell to specify cell fate [20]. 

It is widely accepted that asymmetric cell division is a defining characteristics of stem cells that enables them to self-renew and differentiate. Dysregulated asymmetric division of stem cells is the root cause of tumorigenesis [54]. Clearly, it is particularly important to identify the mechanisms that convert a normal mammalian stem cell into a cancer stem cell. Further investigations of the role of ACBD3 in cancer stem cell differentiation and tumorigenesis may help increase the understanding of the onset of cancer and develop innovative therapies.

## 11. Conclusions

ACBD3 has unusually diverse roles both in normal cell physiology and in diseases. The data covered in this review strongly suggest that a central theme encompassing these diverse activities of ACBD3 is its ability to form distinct scaffoldings on diverse membranous environments. This characteristic diversity, in turn, seems to allow various roles via its interaction with numerous adapters/interacting proteins (Figure 2, Table 1). It still remains to be determined how these distinct membrane micro-domains containing ACBD3 can function without mixing with each other. Due to the recent advancement of big data handling and super-resolution microscopy, there will be interesting opportunities to further understand ACBD3′s diverse intracellular activities with better clarity by a more integrated analysis of basic cell biology and clinical data.

## Figures and Tables

**Figure 1 ijms-20-02028-f001:**
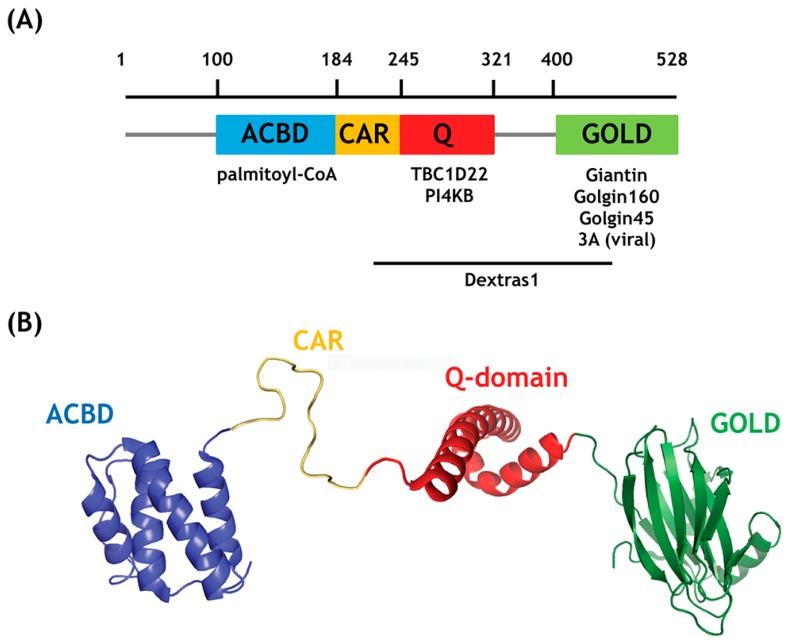
(**A**) Schematic representation of the acyl-CoA-binding domain-containing 3 (ACBD3) domain structure with known binding sites. (ACBD: acyl-CoA binding domain; CAR: charged amino acid region; Q: glutamine-rich domain) (**B**) Pseudoatomic model of ACBD3, based on previously published GOLD (PDB code: 5TDQ) and Q domain (PDB code: 2N72) structures plus a homologous ACBD domain crystal structure from bovine ACBD1 (PDB code: 1ACA), which binds palmitoyl-CoA directly [10,22]. Crystal structure of full length ACBD3 and its ACBD domain are currently not available.

**Figure 2 ijms-20-02028-f002:**
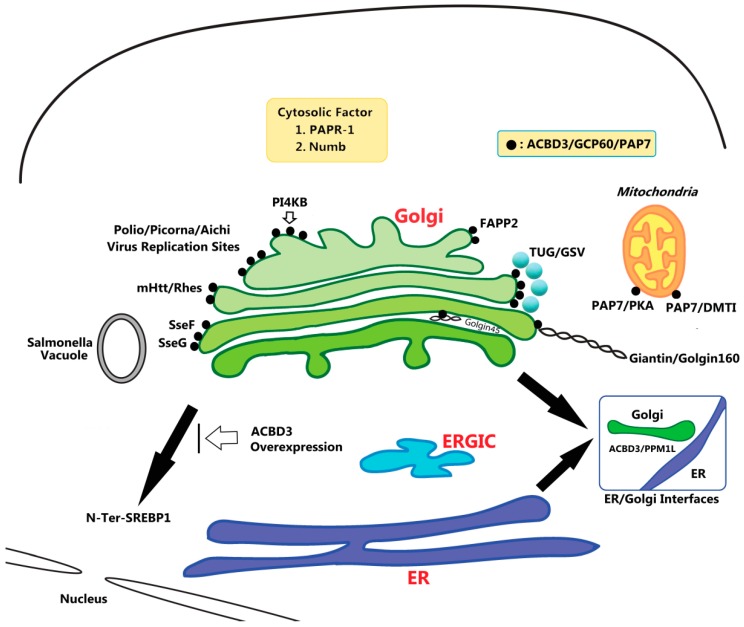
Schematic illustration of diverse roles of ACBD3 and its interaction partners at the Golgi, mitochondria, and cytoplasm in the cells. ACBD3′s physiological roles range from steroidogenesis, neurogenesis, lipid metabolism, cell division, apoptosis, membrane trafficking to proposed roles in various diseases like cancer, diabetes, viral and bacterial replication in host cells, and Huntington’s disease (HD), etc.

**Table 1 ijms-20-02028-t001:** Previously identified ACBD3-interacting proteins.

Protein Names	Molecular Weight	Intracellular Localization	Functions	References
Giantin	372 kDa	Golgi	Golgi organization; ER to Golgi transport	[6]
Golgin160	160 kDa	Golgi	Maintenance of Golgi structure	[7,8]
Golgin45	45 kDa	Golgi and nucleus	Required for normal Golgi structure and for protein transport from the ER through the Golgi to the cell surface	[11]
Numb	72 kDa	plasma membrane, nucleus and cytosol	Plays a role in the determination of cell fates during development	[19]
PI4KB	92 kDa	Golgi, endosome and cytosol	May regulate Golgi disintegration/reorganization during mitosis; Involved in Golgi-to-plasma membrane trafficking	[10,17,21,42,44,45]
TUG	60 kDa	plasma membrane, nucleus and cytosol	Tethering protein that sequesters GLUT4-containing vesicles in the cytoplasm in the absence of insulin. Modulates the amount of GLUT4 that is available at the cell surface. Enhances VCP methylation catalyzed by VCPKMT.	[12]
Htt	35 kDa	cytoskeleton, nucleus, cytosol, ER, Golgi and endosome	Huntingtin is a disease gene linked to Huntington’s disease; May play a role in microtubule-mediated transport or vesicle function	[13]
PBR	18 kDa	mitochondria	Promotes the transport of cholesterol across mitochondrial membranes and may play a role in lipid metabolism	[26]
PRKAR1A	43 kDa	plasma membrane and cytosol	Regulatory subunit of the cAMP-dependent protein kinases involved in cAMP signaling in cells	[26]
3A protein			Nonstructural viral protein required for RNA replication	[14,23,24,42,44,45]
SseF and SseG			Both are (*Salmonella* pathogenicity island 2) SPI-2-encoded effectors which are necessary for the retention of tightly clustered bacterial microcolonies in close proximity to the MTOC and the Golgi	[22]
DMT1	62 kDa	plasma membrane, mitochondria, nucleus lysosome, endosome and Golgi	Important in metal transport, in particular iron	[16]
PARP1	110 kDa	Mitochondria and nucleus	Involved in the base excision repair (BER) pathway, by catalyzing the poly(ADP-ribosyl)ation of a limited number of acceptor proteins involved in chromatin architecture and in DNA metabolism	[20]
SREBP1	120 kDa	ER, nucleus, cytosol and Golgi	Transcriptional activator required for lipid homeostasis	[28]
FAPP2	58 kDa	nucleus and Golgi	Cargo transport protein that is required for apical transport from the Golgi complex; Mediates the non-vesicular transport of glucosylceramide (GlcCer) from the trans-Golgi network (TGN) to the plasma membrane and plays a pivotal role in the synthesis of complex glycosphingolipids	[9]
TBC1D22A	47 kDa	Perinuclear, Golgi	GTPase activating protein (GAP) for Rab33b; membrane trafficking	[14]
PPM1L	41 kDa	ER	Helps regulate ceramide transport from the ER to the Golgi; Acts as a suppressor of the SAPK signaling pathways	[15]

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
