# Peer review of "Acyl-CoA-Binding Domain-Containing 3 (ACBD3; PAP7; GCP60): A Multi-Functional Membrane Domain Organizer"

_ijms, 2019, doi:10.3390/ijms20082028_

Round 1
Reviewer 1 Report
The report by Yue et al. reveals a nice overview on the universal roles of in the Acyl-CoA Binding Domain Containing 3 protein. The authors should still make a few point more clear.
- Through reading the manuscript it does not get clear how the structure of ACBD3 looks like. Is there a structure available of this protein?
- Furthermore a structure or scheme of the protein would be useful to assign the potential binding sites for other proteins, which are described in this report.
- Are there splice variants of ACBD3 which are able to fulfill these different functionalities.
- The authors state the ACBD3 has diverse roles in normal cell and disease cell. Is there knowledge how these different regulation mechanisms in health and disease work? For example in breast cancer cells: Which signaling apthways around ACBD3 are different in healthy and cancerous breast tissue?
Author Response
The report by Yue et al. reveals a nice overview on the universal roles of in the Acyl-CoA Binding Domain Containing 3 protein. The authors should still make a few point more clear.
- Through reading the manuscript it does not get clear how the structure of ACBD3 looks like. Is there a structure available of this protein?
Thank you for pointing out. A schematic representation of ACBD3 domain structure with known binding sites was added as Fig 1. Crystal structure of whole ACBD3 protein is not available currently. To help the readers, a pseudoatomic model, based on published GOLD and Q-domain plus a homologous bovine ACBD domain was added to Fig 1.
- Furthermore a structure or scheme of the protein would be useful to assign the potential binding sites for other proteins, which are described in this report.
See above
- Are there splice variants of ACBD3 which are able to fulfill these different functionalities.
There is no splice variant of ACBD3 that has been identified yet.
- The authors state the ACBD3 has diverse roles in normal cell and disease cell. Is there knowledge how these different regulation mechanisms in health and disease work? For example in breast cancer cells: Which signaling apthways around ACBD3 are different in healthy and cancerous breast tissue?
While ACBD3 has been associated with diabetes (TUG), Huntingtin’s disease (mHtt) and breast cancer, our understanding of these roles still remains at the basic level.
A direct role of ACBD3 in cancer has been seldom studied, although there is a recent paper, which studied the clinical significance of ACBD3 in breast cancer. They found that ACBD3 is up-regulated in breast cancer tissues and that overexpression of ACBD3 is correlated with activation of the β-catenin pathway in human breast cancer samples. However, as stated in the main text, the authors did not thoroughly investigate the exact mechanism of altered β-catenin signaling between healthy and cancerous breast tissue that seems to be induced by ACBD3 overexpression. Biological role of ACBD3 in human breast cancer or other types of cancers need to be further explored.
Reviewer 2 Report
In Acyl-CoA Binding Domain Containing 3 (ACBD3; PAP7; GCP60): a Multi-Functional Membrane Domain Organizer, Yue et al. review the different role of the protein ACBD3, which has revealed as a scaffolding factor usually involved in the formation of membrane associated complexes.
As a general consideration it would helpful to give a brief overview of the ABCD3 structure, defining its different domains (ACBD, CAR, Q, GOLD) and adding a representation of it. This would make the text clearer, since in the current version these domains are mentioned along the text without criteria to write their acronym o their complete full name.
On the other hand is clear from the title that ACBD3 has two other names (PAP7 and GCP60). The authors use commonly the most widespread one, ACBD3, but, in some part of the text, they change the name to PAP7, which is a bit confusing.
The last two sentences at the bottom of the page 3 (lines 68 & 69) needs to be rewrote in a clear way: “GST tagged partia PAP7, fragments…” to GST tagged ACBD3 fragments…
In Page 4, line 87: The protein ACC is mentioned for the first in the text, a explanation of the function of this protein is needed as well as in page 5 line 95 for ERK 1/2
In page 5 line 11 “specialized regions of membrane apposition termed” is not needed.
The second paragraph, in page 8, results redundant after the previous one, where there is an explanation of the relationship between the poliovirus protein 3A and ACBD3. A rearrangement of these two paragraphs is needed.
In the last paragraph of the same page the authors mention ACBD3 is important for bacterial replication and they go further introducing the SseG and SseF, without saying, until several lines later, that the bacteria they are talking about is Salmonella Typhimurium,.
In the discussion again, the same sentence is repeated twice (line 209 and line 212), I suggest for clarity and concision the ideas containing in the paragraphs opening by the same two sentences should be combined into a single paragraph
Author Response
In Acyl-CoA Binding Domain Containing 3 (ACBD3; PAP7; GCP60): a Multi-Functional Membrane Domain Organizer, Yue et al. review the different role of the protein ACBD3, which has revealed as a scaffolding factor usually involved in the formation of membrane associated complexes.
As a general consideration it would helpful to give a brief overview of the ABCD3 structure, defining its different domains (ACBD, CAR, Q, GOLD) and adding a representation of it. This would make the text clearer, since in the current version these domains are mentioned along the text without criteria to write their acronym o their complete full name.
Thank you for pointing out. A schematic representation of ACBD3 domain structure with known binding sites was added to Fig 1. Crystal structure of whole ACBD3 protein is not available currently. To help the readers, a pseudoatomic model, based on published GOLD and Q-domain plus a homologous bovine ACBD domain was added to Fig 1.
On the other hand is clear from the title that ACBD3 has two other names (PAP7 and GCP60). The authors use commonly the most widespread one, ACBD3, but, in some part of the text, they change the name to PAP7, which is a bit confusing.
Thank you for point this out. We fixed these inconsistencies in the main text.
The last two sentences at the bottom of the page 3 (lines 68 & 69) needs to be rewrote in a clear way: “GST tagged partia PAP7, fragments…” to GST tagged ACBD3 fragments…
We fixed this sentence.
In Page 4, line 87: The protein ACC is mentioned for the first in the text, a explanation of the function of this protein is needed as well as in page 5 line 95 for ERK 1/2
Explanations for these terms were added in the text.
In page 5 line 11 “specialized regions of membrane apposition termed” is not needed.
This sentence was fixed, as suggested.
The second paragraph, in page 8, results redundant after the previous one, where there is an explanation of the relationship between the poliovirus protein 3A and ACBD3. A rearrangement of these two paragraphs is needed.
These paragraphs were rearranged and fixed, as suggested.
In the last paragraph of the same page the authors mention ACBD3 is important for bacterial replication and they go further introducing the SseG and SseF, without saying, until several lines later, that the bacteria they are talking about is Salmonella Typhimurium,.
Thank you for pointing out. This was fixed.
In the discussion again, the same sentence is repeated twice (line 209 and line 212), I suggest for clarity and concision the ideas containing in the paragraphs opening by the same two sentences should be combined into a single paragraph
Thank you for pointing out. The two paragraph are now fixed and combined, as suggested.